# Self-Assembly of Au–Ag Alloy Hollow Nanochains for Enhanced Plasmon-Driven Surface-Enhanced Raman Scattering

**DOI:** 10.3390/nano12081244

**Published:** 2022-04-07

**Authors:** Weiyan Liu, Jianwen Zhang, Juan Hou, Haibibu Aziguli, Qiming Zhang, Hu Jiang

**Affiliations:** 1Key Laboratory of Ecophysics, Department of Physics, College of Science, Shihezi University, Xinjiang 832003, China; liuwy19961113@163.com (W.L.); zaz760827@163.com (J.Z.); arzuh2021@shzu.edu.cn (H.A.); zqm990823@163.com (Q.Z.); 2Key Laboratory of Oasis Town and Mountain-basin System Ecology of Xinjiang Bingtuan, Shihezi University, Xinjiang 832003, China

**Keywords:** surface-enhanced Raman spectroscopy (SERS), Au–Ag alloy hollow nanochains, localized surface plasmon resonance (LSPR), FDTD simulations

## Abstract

In this paper, Au–Ag alloy hollow nanochains (HNCs) were successfully prepared by a template-free self-assembly method achieved by partial substitution of ligands. The obtained Au–Ag alloy HNCs exhibit stronger enhancement as surface-enhanced Raman scattering (SERS) substrates than Au–Ag alloy hollow nanoparticles (HNPs) and Au nanochains substrates with an intensity ratio of about 1.3:1:1. Finite difference time domain (FDTD) simulations show that the SERS enhancement of Au–Ag alloy HNCs substrates is produced by a synergistic effect between the plasmon hybridization effect associated with the unique alloy hollow structure and the strong “hot spot” in the interstitial regions of the nanochains.

## 1. Introduction

Nanoplasmonic materials have received much attention due to their intriguing physicochemical properties and promising applications in biomedical [1,2], sequencing [3,4], catalysis [5], and surface-enhanced Raman scattering (SERS) [6]. Plasmonic nanochains are considered excellent SERS substrates due to their large specific surface area, high flexibility, inherent anisotropic morphology, and near-field coupling between nanoparticles leading to enhanced local electromagnetic fields [7]. Compared to isolated nanoparticles, the plasmonic coupling between particles in the nanochains greatly enhances the electric field strength in the interstitial region, which is known as the “hot spot” [8]. The strongly enhanced electric field at the hot spot increases the density of states of photons on the metal surface, which in turn increases the emissivity of the scattering process and enhances Raman scattering [9]. Jia et al. prepared one-dimensional gold nanochains with siloxane surfactants and used them as substrates, which enhanced the SERS response by several orders of magnitude compared to gold nanoparticle substrates [10].

Although significant progress has been made in the preparation and optical studies of self-assembled nanochain structures, most studies have been limited to gold nanoparticles (Au NPs) due to the relatively active chemical nature of Ag and the low chemical stability of the nanoparticle surface [11]. In fact, the local field enhancement factor of Au NPs is much weaker than that of silver nanoparticles (Ag NPs) due to the strong light absorption exhibited by Ag NPs in the visible range based on LSPR [12]. Au–Ag NPs (both alloy and core-shell structures), combining the beneficial plasmonic properties of Au and Ag, have unique physicochemical properties (such as tunable optical properties and good chemical stability) that are usually superior to those of pure metal NPs [13]. In previous work, it was observed that bimetallic NPs, especially silver-based nanostructures, can exhibit excellent optical properties, which can significantly improve the detection sensitivity of SERS-based sensors. For example, Fan et al. investigated the SERS of Au–Ag bimetallic NPs and demonstrated that the SERS enhancement of Au–Ag alloy NPs is higher than that of pure Au or Ag NPs [14]. Mandal et al. also found that Au–Ag NPs exhibited higher SERS enhancement factors than single metals [15]. Moreover, hollow nanomaterials incorporating porous shells and internal voids have better plasmonic properties compared to solid or core-shell structures [16], thanks to a mechanism called plasmon hybridization [17]. Mahmoud et al. revealed by discrete dipole approximation (DDA) and three-dimensional finite difference time domain (FDTD) simulations, that the presence of mutual coupling of internal and external equimolar excitations of hollow nanoparticles can lead to stronger local fields [18]. Therefore, hollow plasmonic nanoparticles have better SERS properties than solid plasmonic nanoparticles.

Usually, self-assembly strategies for nanochains include template-assisted self-assembly and template-free self-assembly. Template-based approaches require additional templates or coupling agents, such as biomolecules [19], polymers [20], or molecular connectors. These attached macromolecules can hinder the access of analytes to hot spots and significantly reduce the sensitivity of SERS analysis using these nanoassemblies as substrates [21]. However, most of the reported processes are still based on template-based methods. Therefore, the use of template-free self-assembly methods for the preparation of nanochains as substrates has great potential in the field of SERS analysis.

In this work, we have successfully prepared Au–Ag alloy HNCs by inducing template-free self-assembly of hollow structures using a partial ligand substitution. We also investigated the application of Au–Ag alloy HNCs as SERS nanoprobes. Using crystalline violet (CV) molecules as probe molecules, the Au–Ag alloy HNCs produced much stronger SERS signals (nearly 1.3 times) than individual Au–Ag alloy HNPs. Moreover, the main characteristic Raman bands of the pesticide thiram molecule can be clearly distinguished, even at concentrations as low as 0.05 ppm, and the detection limits have been reduced to the ultra-low levels obtained in many previous works [22,23], which undoubtedly has great potential for food regulatory applications.

## 2. Materials and Methods

**Chemicals.** Crystalline violet (>99.0%), thiram (97%), silver nitrate (AgNO_3_, 99.99%), β-mercaptoethanol (MEA; HS(CH_2_)_2_OH, 98%) and sodium borohydride (NaBH_4_, 98%) were purchased from Sigma-Aldrich (Shanghai, China). Sodium citrate (SC; C_6_H_5_Na_3_O_7_, 98%) and hydrogen tetrachloroaurate hydrate (HAuCl_4_·4H_2_O, >99.0%) were purchased from Shanghai Sinopharm Chemical Reagent. All chemicals were used as received.

**Synthesis of Au–Ag alloy HNPs****.** Au–Ag alloy HNPs were synthesized according to our previous report [13]. First, citrate-capped Ag NPs (~70 nm) were synthesized as sacrificial templates by the seed-mediated Lee–Meisel method reported by Gu et al. [24]. Then, 20 μL of 0.5 wt% HAuCl_4_ solution with 5 mL of water was boiled for 10 min, and 200 μL of 1 wt% SC solution was injected rapidly and stirred vigorously. After 1 min, 150 μL of pre-prepared Ag NPs were injected, and the solution immediately turned blue. The mixture continued to boil and after stirring for 5 min, the reactor assembly was quickly immersed in an ice water bath and the object was cooled to obtain Au–Ag alloy HNPs.

**Fabrication of Au–Ag alloy HNPs nanochains.** Au–Ag alloy HNCs were prepared by the partial ligand replacement method [25]. The prepared Au–Ag alloy HNPs solution was diluted twice and 1 µL, 5 µL, 10 µL and 15 µL of MEA solution (1.43 mM) were added at room temperature to induce nanoparticle self-assembly. After 72 h, the products were concentrated by centrifugation (6000 rpm, 10 min).

**Materials characterization.** The sample morphology was analyzed by transmission electron microscopy (TEM) and the images were recorded on a Hitachi HT-7700 TEM (Hitachi High-Technologies Corporation, Tokyo, Japan) equipped with a tungsten filament, operating at 100 kV. The JEM-2100F (Japan Electronics Co., Ltd, Tokyo, Japan) was equipped with a built-in energy dispersive X-ray spectrometer (EDS) for testing the STEM- HAADF images and obtaining the EDS elemental maps. The absorption spectra were recorded by a UV–Vis spectrometer (Hitachi U-3900H).

**Surface-enhanced Raman spectroscopy (SERS).** In a typical CV molecular Raman spectroscopy, the SERS substrates were prepared by dropping 5 μL of Au–Ag alloy HNCs concentrate onto a clean hydrophobic silicon wafer. Then, 3 µL of aqueous CV solutions of different concentrations were dropped onto the substrate and dried naturally. Raman spectroscopy was performed using a confocal Raman microscope (Bruker Senterra, Bruker Co., Ltd., Mannheim, Germany) that had a 20 × objective (NA = 0.4) with a power of 1 mW and an excitation wavelength of 532 nm. The exposure time and accumulation numbers were 5 s and 2 times, respectively.

## 3. Results and Discussion

### 3.1. Preparation and Characterization of Au−Ag Alloy HNCs

Figure 1a presents a two-step synthesis strategy for Au−Ag alloy HNCs. Briefly, citrate-covered Ag NPs were synthesized as sacrificial templates by the seed-mediated Lee–Meisel method reported by Gu et al. [24], and Au–Ag alloy HNPs were synthesized according to our previous report [13], and then, self-assembly into nanochains [25]. The size, shape and composition of Au−Ag alloy HNCs were examined by TEM and EDS. Figure 1b shows the overall morphology of the Au–Ag alloy HNCs consisting of uniform Au–Ag alloy HNPs with an average diameter of 70 nm, which have irregular branching. The HRTEM image in Figure 1f shows Au–Ag alloy HNPs with a shell of about 10 nm and lattice spacing values of about 0.235 nm and 0.204 nm. It is consistent with the (111) and (200) planes of the Au–Ag alloy with the face center cubic (fcc) structure reported in the literature as a polycrystalline structure [26,27]. In addition, HRTEM studies indicate that the distance between two adjacent Au–Ag alloy HNPs within the chains is about 1 nm. Elemental mapping diagrams of hollow Au–Ag nanoparticles are given (Figure 1c–e). The uniform distribution of the Ag and Au elements clearly confirms that the obtained nanochains are mixed-phase hollow structures. As shown in Figure 1g, the EDS elemental analysis of the Au−Ag alloy HNCs yielded peaks for gold, silver and sulfur, showing an atomic ratio of Au/Ag equal to 3:7 and MEA molecules associated with the chains.

UV–Vis absorption spectrometer and TEM were used to carefully monitor the self-assembly process of the alloy nanochains. As shown in Figure 2a, the initial Au–Ag alloy HNPs solution has a distinct narrow absorption peak at about 580 nm, which can be attributed to the typical LSPR of the nanoparticles [13]. For a relatively low addition of 1.43 mM MEA, a longitudinal plasma excitonic absorption peak located at around 860 nm started to appear in the absorption spectrum after 72 h (Figure 2b). With increasing addition, the intensity of the absorption peak at 580 nm decreases significantly while the intensity of the longitudinal plasmon peak gradually increases and slightly redshifts, which is related to the progressive aggregation of the nanoparticles into chains and branched networks [20,28]. TEM images corresponding to the above spectral changes show the presence of short chains and discrete nanoparticles at relatively low additions of MEA (Figure 2c,d), and with increasing MEA, the length of short chains starts to increase, with the appearance of branching and interconnected complex chains (Figure 2e), as well as isotropic aggregates that twist after additions above 15 μL (Figure 2f). When the charged citrate capping ligand partial is partially replaced by neutral MEA, chain formation is triggered due to the intermolecular interaction between adjacent MEA hydroxyalkyl chains and specific interactions of MEA head groups with a subset of the exposed crystallographic surfaces of the twinned Au nanoparticles. Once all the citrate ions are substituted, the electrostatic repulsive force is reduced to zero and isotropic bulk aggregation occurs. Thus, the presence of a mixed citrate/MEA stabilization shell is responsible for the generation of Au–Ag alloy HNCs, which is consistent with the results of the study. The above results and discussion indeed confirm that well-defined Au–Ag alloy HNCs can be simply prepared by adding MEA to the alloy nanoparticle solution without any structure-directed additives and complex reaction conditions.

In order to elucidate the mechanism of chain self-assembly, time-lapse spectra of the samples were recorded at the addition of 10 µL of MEA. Figure 3a shows the time-lapse UV–Vis spectra of the mixed solution after the addition of 10 µL of MEA, and Figure 3b is a partially enlarged view of Figure 3a. Within 1 h of MEA addition, the spectrum shows a slight redshift of the absorption peak at 580 nm, which may be due to the ongoing nano-assembly. After 6 h, a low-energy shoulder peak centered at 850 nm appears. The low-energy shoulder peak gradually redshifted with increasing time, and no further changes were observed at 72 h. The corresponding TEM images showed that the nanoparticles were initially mainly separated after the addition of MEA (Figure 3c). Within 6 h, self-assembly of short chains, usually consisting of a few nanoparticles, occurs (Figure 3d). With time, the length of the nanochains increases, and more and more bifurcations and closed loops are formed, resulting in an extended network of interconnected nanoparticle chains within 72 h (Figure 3e). These superstructures remained unchanged even when the incubation time was extended (e.g., two weeks) (Figure 3f). These results indicate that Au–Ag alloy HNCs with different morphologies and lengths can be obtained by carefully controlling the amount of MEA added and the reaction time.

### 3.2. SERS Effect of Au–Ag Alloy HNCs

The SERS properties of Au–Ag alloy HNCs were investigated by using crystal violet (CV) as a probe molecule. For comparison, monodisperse Au–Ag alloy HNPs, 20 nm Au NPs and the corresponding NCs substrates were chosen as reference SERS substrates. The SERS spectra of the 10^−6^ M CV molecules adsorbed on the above four SERS substrates are shown in Figure 4a. Additionally, the SERS spectra are the results obtained by averaging the Raman spectra taken several times at different locations on the substrate. The main characteristic Raman peaks of the CV molecules, such as 1176, 1375 and 1620 cm^−1^, can be clearly detected for all substrates. The SERS activity of Au–Ag alloy HNCs is better than that of Au–Ag alloy NPs, and Au NCs are better than that of Au NPs, indicating that the self-assembly chain plays a very important role in SERS enhancement. Due to the formation of HNPs junctions, HNCs can provide a number of plasmonic hot spots, which greatly enhance the electric field strength in the interstitial region and provide an extremely strong local electric field at the junctions, resulting in a more intense SERS signal. In addition, the SERS performance of the Au–Ag alloy HNCs substrate was improved by nearly 1.3 times over that of the Au–Ag alloy HNPs substrate, indicating that the plasmon hybridization effect due to the unique hollow structure associated with the two surfaces provided higher SERS enhancement. Then, the SERS performance of Au–Ag alloy HNCs substrates was examined using different concentrations of CV from 10^−5^ M to 10^−8^ M (Figure 4b). The main characteristic peaks were also clearly identified when the concentration was reduced to 10^−8^ M. The ultra-sensitive detection limit of Au–Ag alloy HNCs was close to that required for single-molecule detection (~nM). The comparative results show that the Au–Ag alloy HNCs have a strong competitive advantage for SERS applications.

The excellent performance of Au–Ag alloy HNCs is mainly attributed to the plasmon hybridization effect and “hot spot” effect, both of which are related to electromagnetic enhancement. Therefore, we further quantitatively simulated the electromagnetic distribution of Au NPs, Au–Ag alloy HNPs, Au NCs and Au–Ag alloy HNCs using the FDTD method. The geometrical parameters of the model structures were assumed to be close to those of the actual ones. Namely, the diameters of Au NP and Au–Ag alloy HNP were set to 20 and 70 nm, respectively, and the shell thickness of Au–Ag alloy HNPs was set to 10 nm. The Au–Ag alloy HNCs were set as 1D periodic structures and the gap between Au NPs was fixed at 1 nm. The dielectric constant of Au was taken from Johnson and Christy’s measurements with polynomial fitting [29], and the dielectric constant of Au: Ag (3:7) alloy was adopted from an analytic model proposed by Michel Meunier [30]. The FDTD simulation of local electric intensity can be seen in Figure 5, where the ratio of the maximum local electric intensity of Au–Ag alloy HNCs and Au NCs matches well with the ratio of the SERS intensity of the CV on Au–Ag alloy HNCs and Au NCs (1.3:1). The ratio of the maximum local electric intensity of Au NCs to Au–Ag alloy HNPs is about 5:1, however, the SERS intensity of CV on Au NCs and Au–Ag alloy HNPs are essentially the same (Figure 4a). Note that the maximum localized electrical intensity only represents the SERS enhancement at a specific position. For the measurement of the ensemble SERS, any molecules distributed on the surface of the nanostructure must be considered. Therefore, the local electric field intensity should be spatially averaged. As shown in Figure 5b,c, the internal electric intensity of Au–Ag alloy HNPs is stronger than that of Au NCs due to the coupling of the LSPR at the internal and external surfaces. Thus, the spatially averaged electric field strengths of both are essentially the same, which matches well with the experimental results. The above analysis demonstrates that, in addition to the hot spot effect, the plasmon hybridization effect also plays a very important role in the SERS enhancement.

To evaluate the potential of Au–Ag alloy HNCs as SERS materials, we measured the Raman spectroscopy of thiram, a pesticide that often remains on fruits and vegetables. Figure 6a demonstrates the Raman spectra of different concentrations of thiram solutions. The characteristic Raman peak of thiram at 1380 cm^−1^ (CH_3_ symmetric rocking, C–N stretching) can be detected even down to 0.05 ppm, which is sufficient to prove the excellent detection ability of the Au–Ag alloy HNCs substrate. Figure 6b illustrates that the Raman intensity of thiram maintains a great linear correlation (R^2^ = 0.969) over the concentration range from 0.05 ppm to 10 ppm, enabling the qualitative and quantitative detection of thiram at ultra-low concentrations. All the above findings suggest that the Au–Ag alloy HNCs substrates have excellent SERS performance.

## 4. Conclusions

In conclusion, Au–Ag alloy HNCs were prepared simply and rapidly by using a template-free self-assembly method. The controlled morphology and length of Au–Ag alloy HNCs were obtained by carefully tuning the controlled MEA addition and reaction time. Excellent SERS properties of Au–Ag alloy HNCs substrates were confirmed by Raman experiments on CV as well as thiram molecules. Both experimental and computational studies have demonstrated that the great SERS response of Au–Ag alloy HNCs substrate is generated by the synergistic effect between the plasmon hybridization effect associated with the unique alloy hollow structure and the strong “hot spot” in the nanochains interstitial region. These results provide valuable information on design strategies for the design of nanoplasmonic SERS substrates.

## Figures and Tables

**Figure 1 nanomaterials-12-01244-f001:**
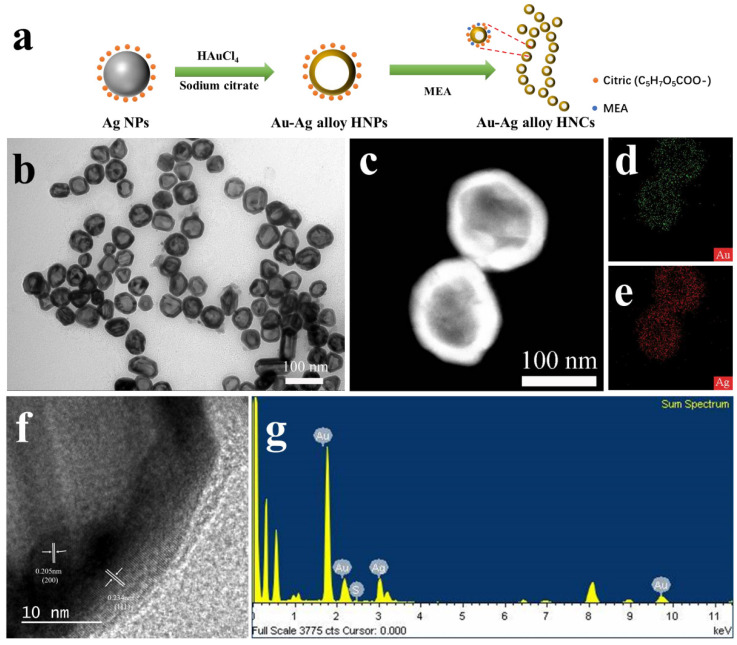
Synthesis of Au–Ag alloy HNCs by a template-free self-assembly method. (**a**) A scheme illustrating the synthesis route of Au–Ag alloy HNCs. (**b**) TEM image of Au–Ag alloy HNCs. (**c**–**e**) HAADF and EDS mapping images of Au–Ag alloy HNCs, respectively. (**f**) HRTEM image of Au–Ag alloy HNCs. (**g**) EDS elemental analysis of the Au−Ag alloy HNCs.

**Figure 2 nanomaterials-12-01244-f002:**
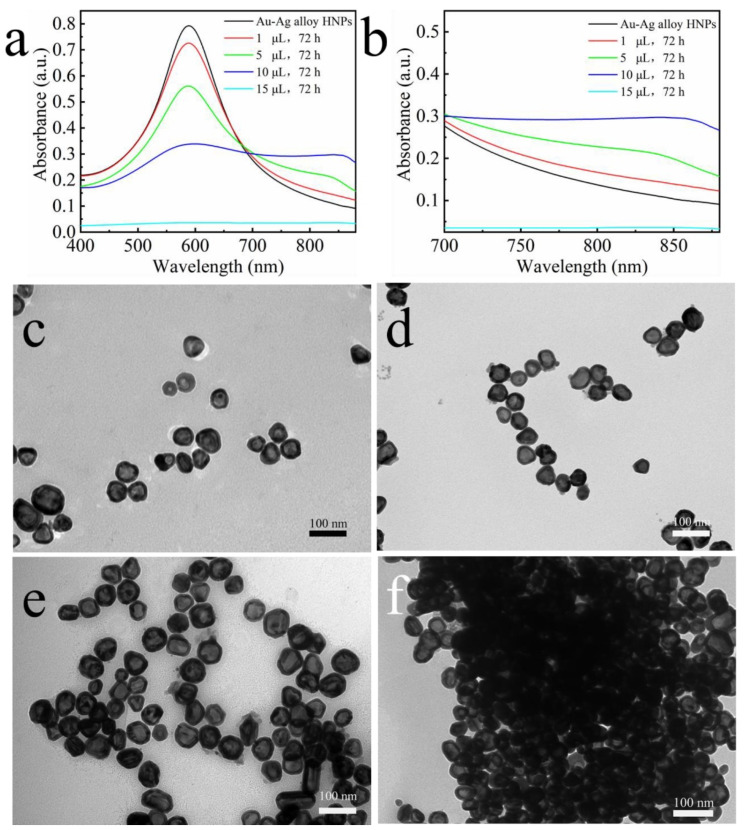
(**a**,**b**) UV–Vis spectral recordings of citrate-stabilized Au–Ag HNPs after 72 h of MEA addition. (**c**–**f**) Corresponding TEM images at different MEA additions (**c**) 1 µL (**d**) 5 µL (**e**) 10 µL (**f**) 15 µL.

**Figure 3 nanomaterials-12-01244-f003:**
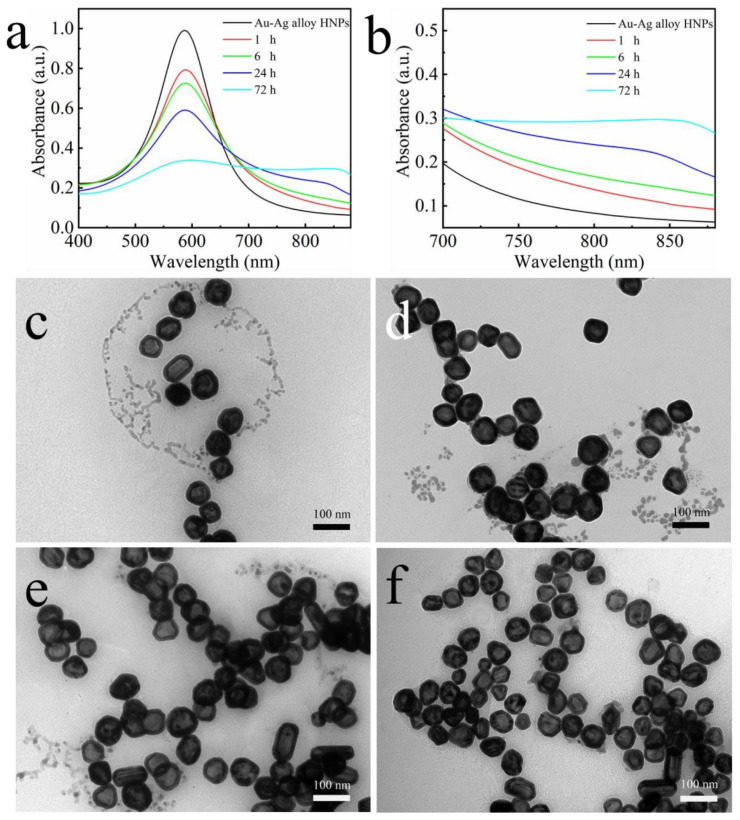
(**a**,**b**) Time-dependent UV–Vis spectra of Au–Ag alloy HNCs sol recorded at various time intervals after the addition of 1.43 mM of MEA. (**c**–**f**) Corresponding TEM images at (**c**) 1 h (**d**) 6 h (**e**) 72 h (**f**) two weeks.

**Figure 4 nanomaterials-12-01244-f004:**
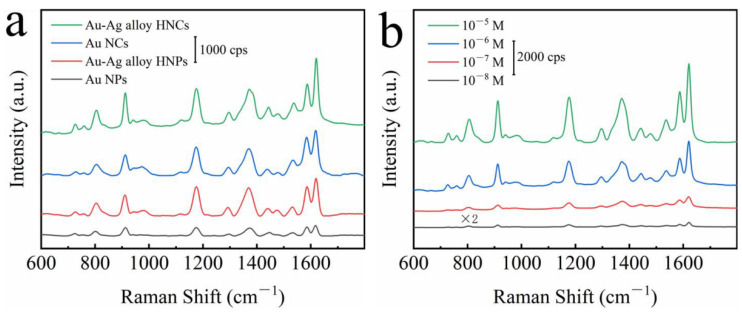
(**a**) The SERS spectra of 10^−6^ M CV adsorbed on Au NPs, Au NCs, Au–Ag alloy HNPs and Au–Ag alloy HNCs. (**b**) SERS spectra of CV molecules with different concentrations (10^−5^−10^−8^ M) adsorbed on Au–Ag alloy HNCs.

**Figure 5 nanomaterials-12-01244-f005:**
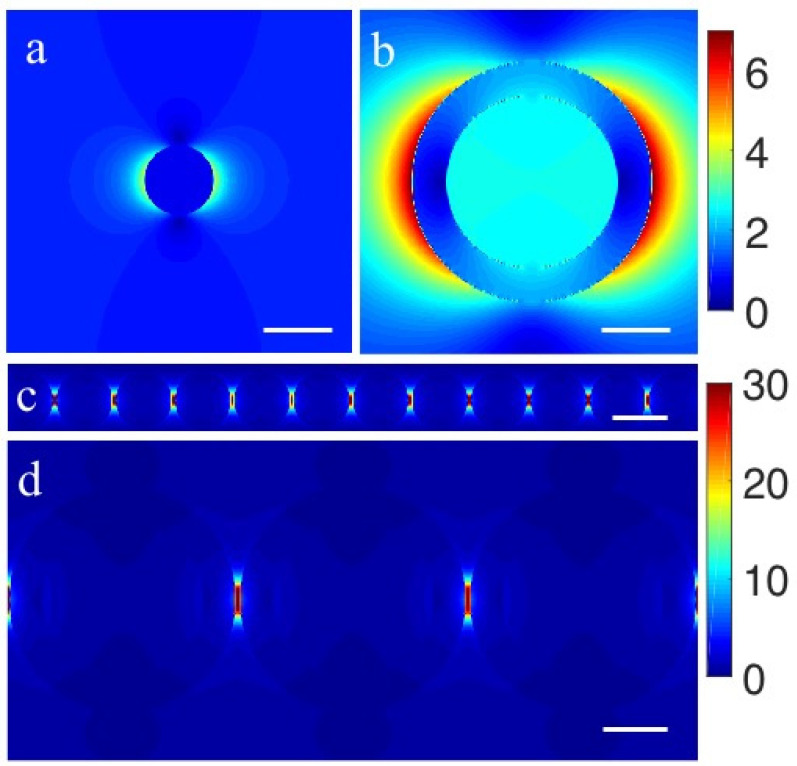
Local electric field distributions of (**a**) Au NPs, (**b**) Au–Ag alloy HNPs, (**c**) Au NCs and (**d**) Au–Ag alloy HNCs, respectively. All scale bars are 20 nm.

**Figure 6 nanomaterials-12-01244-f006:**
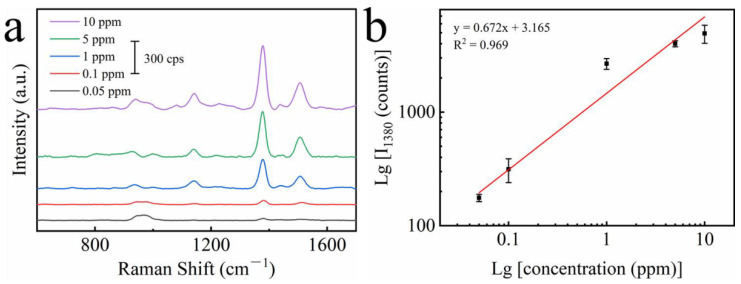
(**a**) Raman spectra of thiram with various concentrations on Au–Ag alloy HNCs substrate. (**b**) Relationship between the intensity of the SERS peak at 1380 cm^−1^ and thiram concentration; the error bar is based on four parallel SERS spectra.

## Data Availability

The data presented in this study are available on request from the corresponding author.

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
