# Peer review of "Self-Assembly of Au–Ag Alloy Hollow Nanochains for Enhanced Plasmon-Driven Surface-Enhanced Raman Scattering"

_nanomaterials, 2022, doi:10.3390/nano12081244_

Round 1

Reviewer 1 Report

The manuscript by Weiyan Liu et. al. titled “Self-Assembly of Au-Ag alloy hollow nanochains for enhanced plasmon-driven Surface-Enhanced Raman Scattering” describes simple procedure of hybrid Au-Ag nanochain preparation which has beneficial effect on enhancement of Raman scattering. The authors make an interesting observation that the addition of small amount of MEA results in aggregation of hollow nanoparticles into chains which eventually lead to more “hot spots” due to touching surfaces. The effect is demonstrated on CV and further on thiram suggesting very low level of detection sensitivity down to 0.05 ppm.

The following are specific concerns that need to be addressed:

  • 2 line 66 claims that nearly 1.3 orders of magnitude while in fact the paper only demonstrates that the signals are 1.3 times larger (Figure 4).
  • It is unclear whether the enhancement effect is due to hybrid nature of nanochains or simply due to the fact that there are more enhancing hot-spots within the excitation area.
  • Comparative results of enhancement (Figure 4 a) should be averaged over several spots and reported as mean values with corresponding standard deviations. Comparing one spectrum is not sufficient to evaluate the efficiency of the overall substrate performance.

Reviewer 2 Report

In their work Liu and colleagues present a plasmonic substrate platform made of Au-Ag alloy hollow nanochains to enhance surface-enhanced Raman scattering signal.  To create these nanochains, they make use of a template-free self-assembly method achieved by partial substitution of ligands. The obtained Au-Ag alloy nanochains exhibit stronger enhancement as surface-enhanced Raman scattering substrates than Au-Ag alloy nanoparticles and Au nanochains substrates with an intensity ratio of about 1.3:1:1. FDTD simulations show that the SERS enhancement of Au-Ag alloy nanochains substrates is produced by a synergistic effect between the plasmon hybridization effect associated with the unique alloy hollow structure and the strong "hot spot" in the interstitial regions of the nanochains. The work is interesting and well written although I have some minor points I would like to ask the Authors to address before giving a final opinion.

  1. It is not clear why the Authors compare the performance of the HNCs with the Au-Ag alloy HNPs and Au HNCs, without considering Ag HNCs, which are known to be very effective in enhancing SERS signals (see for instance Scientific Reports Scientific Reports 8:12652 (2018)). A comparison with a pure Ag HNCs system might be very insightful and the Authors should also comment on this aspect, since this comparison is totally missing.
  2. The justification of using Au-Ag alloy instead of pure Au or Ag HNCs and/or HCPs is not clear to me. Can the Authors add a sentence or two to support their choice to use this alloy instead of pure Ag or Au?
  3. The Authors stress that this type of substrates can be used for enhancing SERS response. Can these structures be used also for enhancing SERS in sequencing applications (see for instance Nano Letters 19 (2), 722-731 (2019) or Nature Communications 10: 5321 (2019)).

Reviewer 3 Report

This paper is of general interest for applications of Au-Ag alloy hollow nanochains, Au-Ag HNCs, for various types of SERS detection.  My main concern is in the comparison of Au-Ag HNCs with respect to Au-Ag HNPs. Is there any real difference or advantage for hollow nanochains over hollow Au-Ag nanoparticles? In other words, is 1.3:1 ratio just determined by experimental conditions. The authors should consider the following comments in a revision:

  1. The Au-Ag HNCs do not show well-developed plasmon extinction UV-VIS spectra in Figures 2b and 3b. On the other hand, Au-Ag HNPs grown by galvanic replacement have shown much better-behaved UV-VIS spectra with broad peaks between 600-800 nm (Sci Rep. 2017 Jan 30;7:41419. doi: 10.1038/srep41419) for various Au/Ag ratios. Also, well-developed UV-VIS spectra  are shown in reference 20 ( Fig.2)for Au NPs chains grown using the same MEA preparation method. How do the mole ratio of MEA/Au-Ag HNPs described on page 2 line 87 compare with the r values in reference 20? Indeed, have the authors studied a wider range of variables in the preparation method including r ratios and temperatures on the plasmon extinction UV-VIS spectra.
  2. Furthermore, what is the effect of the Au/Ag ratio on the UV-VIS spectrum studied with the MEA preparation method? How does the SERS enhancement results obtained for Au-Ag HNCs versus the Au-Ag HNPs depend on the Au/Ag ratios and the preparation details?
  3. Some discussion on page 10 is confusing. On line 189, “Ag-Ag alloy HNPs” should replace Ag-Ag alloy HNCs which is used twice. Also, on line 222 of this page there is no Au HNCs in Figure 5ab. This should be Au NPs. More to the point, can the electric field distribution for Au-Ag HNCs be compared with Au-Ag HNPs for similar sized structures?

Reviewer 4 Report

The paper Self-Assembly of Au-Ag Alloy... by Liu et al deals out the preparation of Au-Ag alloy hollow nanochains, their characterization including plasmonics properties. The paper is well written , well organized and the findings are described in an excellent way.

My only criticism is about the brief bibliography, especially as regards the impact of nanochains in the biomedical field. I suggest the authors to cite the following article which could very well complete the bibliography:

Exploiting gold nanoparticles for diagnosis and cancer treatments

M D’Acunto, P Cioni, E Gabellieri, G Presciuttini Nanotechnology 32 (19), 192001   After  integrating the bibliography I suggest that the article has all the criteria to be published on nanomaterials.

Round 2

Reviewer 1 Report

The authors have responded to my concerns with great detail. However, my original comments stand as stated. The response to my Comment 2 should be incorporated into the discussion of SERS results stating that the authors "believe" the SERS effect is in fact a synergistic effect as they detailed in their response. Comment 3: the description of the spectral averaging procedure over several spots on the substrate should be incorporated in the paper as the authors claim in their response that the spectra presented in Fig. 4 are averaged SERS spectra.  

Reviewer 3 Report

The authors have answered my questions and made the necessary correction to the text.

Round 3

Reviewer 1 Report

The authors have addressed all my concerns. Therefore, I deem the manuscript suitable for publication.